# Pomegranate Cultivars with Diverse Origins Exhibit Strong Resistance to Anthracnose Fruit Rot Caused by *Colletotrichum gloeosporioides*, a Major Disease in Southeast United States

Alexander Schaller [1], John M. Chater [2], Gary E. Vallad [3], Jeff Moersfelder [4], Claire Heinitz [4] and Zhanao Deng [1,*]

1 Department of Environmental Horticulture, Gulf Coast Research and Education Center, University of Florida, IFAS, 14625 County Road 672, Wimauma, FL 33598, USA; aschaller@ufl.edu
2 Department of Horticultural Sciences, Citrus Research and Education Center, University of Florida, IFAS, 700 Experiment Station Rd., Lake Alfred, FL 33850, USA; jchater@ufl.edu
3 Department of Plant Pathology, Gulf Coast Research and Education Center, University of Florida, IFAS, 14625 County Road 672, Wimauma, FL 33598, USA; gvallad@ufl.edu
4 National Clonal Germplasm Repository, USDA-ARS, University of California, Davis, CA 95616, USA; jeff.moersfelder@usda.gov (J.M.); claire.heinitz@usda.gov (C.H.)
* Correspondence: zdeng@ufl.edu

**Abstract:** Pomegranate, a pivotal fruit that is well recognized globally and a rapidly emerging crop in the southeastern United States and other subtropical regions, faces a formidable challenge from *Colletotrichum* spp., a fungal pathogen causing anthracnose fruit rot, which leads to severe to complete premature fruit drop. The development and use of disease-resistant cultivars are considered the most cost-effective and sustainable approach to managing this disease. Identifying sources of resistance is essential for developing new cultivars with improved resistance to this disease. This project aimed to expand the scope of evaluation through a 2-year field study in central Florida, examining fruit from 35 cultivars from diverse origins using both artificial inoculation at the petal dehiscent stage and natural infection. Lesion size on the fruit was measured during the growing season in a field setting. Subsequently, seven cultivars were selected for further testing by inoculating detached mature fruit and measuring lesion size to confirm observed resistance and determine the correlation between resistance observed in planta in the field and on detached fruit in the laboratory. The field study revealed significant genetic differences among pomegranate cultivars in susceptibility to naturally occurring and induced anthracnose fruit rot and classified cultivars into five resistance or susceptibility classes. Five cultivars that originated from different regions of the world, including 'Azadi', showed consistent resistance to anthracnose fruit rot in the field. Resistance remained strong on detached mature fruit. A strong positive correlation existed between resistance levels on in-planta fruit and on detached mature fruit, suggesting a possible simple, efficient approach to screening breeding populations for anthracnose fruit rot resistance in pomegranate. These findings represent an important step toward developing new anthracnose-resistant cultivars and understanding and improving disease resistance in this increasingly important fruit crop in the world.

**Keywords:** *Punica granatum*; disease resistance; fungal pathogen; *Colletotrichum*; breeding; germplasm; genetic diversity

## 1. Introduction

The pomegranate (*Punica granatum* L.) is a subtropical fruit tree with a long history of cultivation across the world. Commercial pomegranate orchards can be found in the Middle East and Caucasus region, North and tropical Africa, the Indian subcontinent, Central Asia, Southeast Asia, the Mediterranean Basin, North and South America, and Australia [1]. World pomegranate production is estimated to be well above 300,000 ha [2]

and an estimated 3 million metric tons annually as of 2017 [3]. In recent years, consumer demand for pomegranate has been increasing worldwide [4–6] due to its multiple health benefits [7,8], including strong antimicrobial and antiviral activities [9]. Research on increasing the production and supply of quality pomegranate fruit is much needed to meet consumer demand for this superfood. Destructive diseases have been the most important constraint for successful commercial production of pomegranate in many countries.

Spanish settlers introduced pomegranates to the southern United States and Mexico [10,11] when they first colonized North America in the 1700s. Current pomegranate production in the United States remains relatively small compared to other tree fruit crops, with 12,736 ha of production, 98% of which is in California as of 2017 [12]. The crop has garnered increasing interest in the United States [13,14], including in Florida [15,16]. Pomegranates have been cultivated in Florida since their arrival with Spanish settlers, but commercial production remains very limited. Early research in Florida revealed that pomegranates can grow well in the state, but fruit production is threatened by the high incidence of fungal diseases, particularly anthracnose fruit rot caused by *Colletotrichum* species [17–19]. Anthracnose in pomegranates appears on leaves as small circular leaf spots with yellow halos and on fruit as brown lesions that progress through the fruit causing external and internal rot. In many cases, disease pressure is so intense that the fruit succumbs to rot well before maturity, resulting in up to 100% fruit loss (Figure 1).

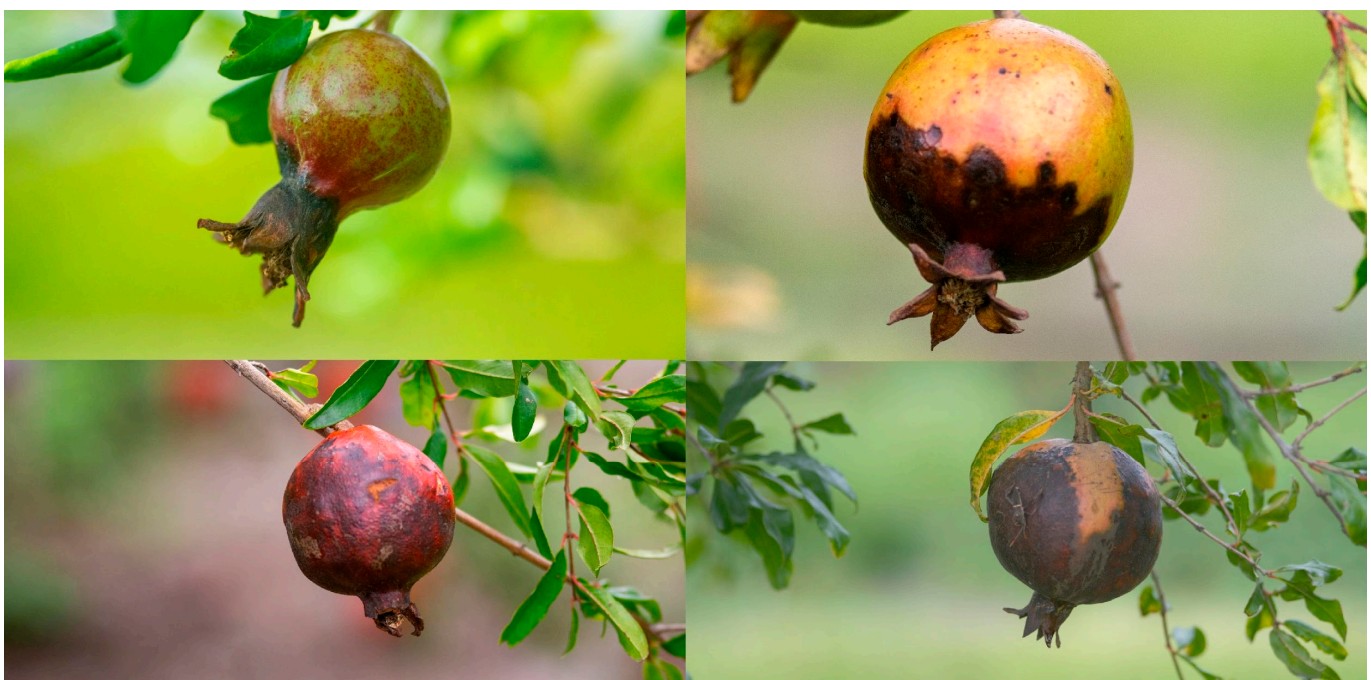

**Figure 1.** Anthracnose symptoms on pomegranate fruit. The symptoms typically begin at the calyx and work their way up until the fruit is completely rotted.

*Colletotrichum* has been reported worldwide in pomegranates [20–25] and thrives in the higher temperature and humidity in subtropical environments such as Florida, Southeast Asia, and India. While some fungicides have been researched and approved for fungal control in pomegranates in multiple countries [26–29], disease-resistant cultivars are considered essential for commercial production in subtropical regions including Florida. Use of disease-resistant cultivars represents a much more sustainable and cost-effective option for managing pomegranate diseases not only in Florida but also globally.

Identifying sources of disease resistance is the first and most critical step for developing new cultivars with greater disease resistance. With hundreds of cultivars worldwide, there is a vast range of potential diversity for screening for disease resistance [30,31]. However,

very limited research has been published on pomegranate cultivars that are resistant to *Colletotrichum* spp., and of the cultivars that have been screened, very few show strong resistance. Joshi et al. (2014) [32] screened cultivars for *Colletotrichum gloeosporioides* fruit rot resistance and found that two popular Indian cultivars, 'Arakta' and 'Bhagwa', were susceptible. However, two local cultivars, 'Yarcud Local' and 'Bedana', showed resistance to *C. gloeosporioides* isolates. Jayalakshmi et al. (2015) [33] investigated nineteen pomegranate cultivars for resistance to *C. gloeosporioides* using detached leaves but found no resistance, with 'Arakta', 'Ganesh', and 'Kesar' showing a higher level of susceptibility. Yu et al. 2018 [34] also reported 'Arakta' and 'Bhagwa' as being highly susceptible to *C. gloeosporioides* in a detached leaf assay, while a local Florida cultivar 'Cedar Key Sunset' exhibited moderate resistance. These previous results indicated that resistance to *Colletotrichum* might exist within the pomegranate germplasm. Our hypothesis for this study was that by expanding screening efforts to include a larger number of cultivars from a diverse background, more useful sources of resistance to *Colletotrichum* could be identified for pomegranate breeding.

The objectives of this study were to (1) evaluate a subset of the USDA pomegranate germplasm collection and a group of heritage Florida cultivars for resistance to fruit rot caused by *C. gloeosporioides* in the 'real world' under natural disease pressure, (2) confirm identified resistance through artificial inoculation of the pathogen, and (3) determine the correlation between the resistance levels observed on in-planta fruit in the field and on detached mature fruit in the laboratory. The evaluation of these cultivars for resistance to this highly destructive fungal disease is important for developing new cultivars with greater disease resistance and expanding our understanding of disease resistance in pomegranate. The results revealed remarkable genetic diversity among the cultivars in terms of susceptibility to the disease and identified five cultivars with strong resistance. Interestingly and unexpectedly, these sources of resistance have a diverse origin in the world. Strong fruit rot resistance was observed to be expressed on detached mature fruit. These new findings can play an important role not only for breeding new pomegranate cultivars with resistance to anthracnose fruit rot but also for managing other important diseases in pomegranate production in the world.

## 2. Materials and Methods

### 2.1. Experimental Site and Pomegranate Cultivars

The experiments were conducted over two years, from 2021 and 2022, at the University of Florida's Gulf Coast Research and Education Center in central Florida. The region's climate is characterized by hot, humid summers with frequent rains from May through September and warm, dry winters (Figure 2). This weather pattern creates a conducive environment for fungal growth during late spring into summer when pomegranate trees are producing flowers and young fruit.

The experimental pomegranate orchard was established in 2015 and was grouped by cultivar with two or three plants per cultivar. Before each evaluation season, the plants were subjected to defoliation utilizing ethephon to encourage earlier blooming. No fungicides were applied during the two growing seasons to ensure high pressure of natural fruit rot disease. In the 2022 season, a series of freezes in January and February caused production issues (few flowers) in some cultivars that limited the number of fruit available for evaluation. Thirty-five pomegranate cultivars were included in 2021, and 27 cultivars in 2022, due to some trees not producing enough fruit for evaluation during that growing season (Table 1). These cultivars originated from five different regions of the world, including the southeastern United States (Florida and Georgia), western United States (California), Turkmenistan and adjacent region, the former Soviet Union, and India [35–37]. The United States Department of Agriculture (USDA) germplasm accession numbers (DPUN) for all cultivars are included in Supplementary Table S1.

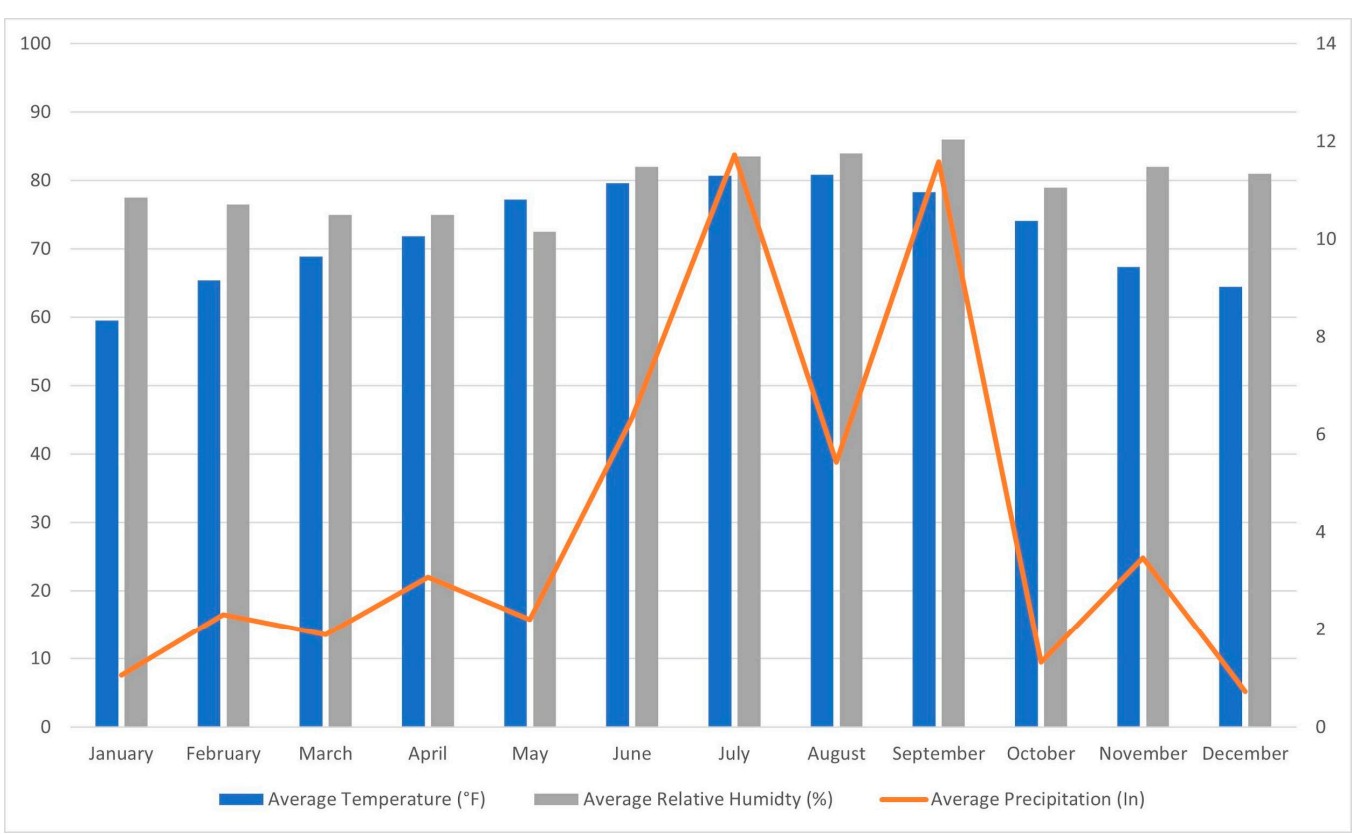

**Figure 2.** The average temperature, relative humidity, and precipitation in central Florida where the experiments were conducted. Data were collected from Florida Automated Weather Network (FAWN).

**Table 1.** Average fruit rot severity rating for fruit that were naturally infected or artificially inoculated for 35 pomegranate cultivars. The disease severity rating is on a scale of 0–6, with 6 being the most susceptible fruit. The top three most susceptible and most resistant cultivars for each treatment are bolded. The standard error is included for each cultivar.

| Cultivar | Artificial Inoculation 2021 | | Natural Infection 2021 | | Artificial Inoculation 2022 | | Natural Infection 2022 | |
|---|---|---|---|---|---|---|---|---|
| Afganski | 2.1 ± 0.71 | bc | 2.3 ± 0.37 | a–i | 4.0 ± 0.38 | abcd | 3.7 ± 0.37 | a–f |
| Al-Sirin-Nar | 3.0 ± 0.52 | abc | 4.0 ± 0.29 | abcd | **5.3 ± 0.24** | **ab** | **5.0 ± 0.20** | **ab** |
| Ambrosia | **4.0 ± 0.47** | **ab** | 3.7 ± 0.28 | a–e | 4.5 ± 0.27 | abc | 4.9 ± 0.18 | abc |
| Angel Red | 1.3 ± 0.56 | bc | 1.8 ± 0.48 | c–i | | | 3.3 ± 0.52 | a–f |
| Apseronski Krasnyj | 3.6 ± 0.18 | ab | 3.9 ± 0.33 | abcd | | | | |
| Arakta | 1.4 ± 0.98 | bc | **0.4 ± 0.56** | **hi** | 0.8 ± 0.68 | gh | **0.8 ± 0.14** | **gh** |
| Azadi | **0.1 ± 0.12** | **c** | **0.1 ± 0.08** | **i** | **0.1 ± 0.07** | **h** | **0.3 ± 0.29** | **h** |
| Bala Miursal | 2.1 ± 0.63 | bc | 2.8 ± 0.46 | a–g | 1.4 ± 0.41 | efgh | 2.3 ± 0.29 | defg |
| Christina | 1.6 ± 0.94 | bc | 1.4 ± 0.45 | d–i | **0.2 ± 0.20** | **h** | 2.8 ± 0.95 | c–g |
| Cranberry | 1.2 ± 0.68 | bc | 3.8 ± 0.30 | abcd | 4.3 ± 1.09 | abcd | 1.6 ± 0.46 | fgh |
| Desertnyi | 3.3 ± 0.29 | ab | **4.4 ± 0.17** | **ab** | 4.9 ± 0.18 | ab | 4.3 ± 0.26 | abcd |
| Don Somner South | 1.1 ± 0.71 | bc | 1.9 ± 0.62 | c–i | 3.1 ± 0.33 | cde | 3.9 ± 0.43 | a–e |
| Eve | 2.9 ± 0.64 | abc | **4.3 ± 0.38** | **abc** | **5.5 ± 0.11** | **a** | **5.2 ± 0.24** | **ab** |

**Table 1.** *Cont.*

| Cultivar | Artificial Inoculation 2021 | | Natural Infection 2021 | | Artificial Inoculation 2022 | | Natural Infection 2022 | |
|---|---|---|---|---|---|---|---|---|
| Eversweet | 2.0 ± 1.39 | bc | 1.2 ± 0.54 | e–i | **0.7 ± 0.45** | **h** | 1.6 ± 0.51 | fgh |
| Fleischman | **1.0 ± 0.52** | **bc** | **0.4 ± 0.20** | **ghi** | 1.1 ± 0.60 | fgh | **0.5 ± 0.18** | **gh** |
| Gainey Sweet | 1.2 ± 0.64 | bc | 0.7 ± 0.38 | ghi | | | | |
| Girkanets | 1.7 ± 0.65 | bc | 2.2 ± 0.49 | b–i | 2.3 ± 0.41 | defg | 2.6 ± 0.35 | defg |
| Gissarskii Rozovyi | 1.3 ± 0.58 | bc | 1.3 ± 0.39 | d–i | | | | |
| Grenada | 1.6 ± 0.68 | bc | 2.8 ± 0.45 | a–g | 3.1 ± 0.58 | cde | 3.9 ± 0.27 | a–e |
| Jimmy Roppe | 1.4 ± 0.59 | bc | 0.6 ± 0.21 | ghi | 2.0 ± 0.60 | d–h | 2.0 ± 0.66 | d–h |
| Kaim-anor | 3.2 ± 0.64 | ab | 3.3 ± 0.51 | a–f | 4.5 ± 0.32 | abc | 4.8 ± 0.26 | abc |
| Kazake | **5.2 ± 0.38** | **a** | **4.6 ± 0.00** | **a** | **5.3 ± 0.16** | **ab** | **5.3 ± 0.12** | **a** |
| Larkin | 3.8 ± 0.88 | ab | 2.5 ± 0.15 | a–h | 2.7 ± 0.45 | def | 2.6 ± 0.45 | defg |
| Medovyi Vahsha | 3.1 ± 0.47 | abc | 1.8 ± 0.59 | c–i | | | | |
| Molla Nepes | **4.4 ± 0.00** | **ab** | 1.2 ± 0.39 | e–i | | | | |
| Nikitski Ranni | 3.4 ± 0.52 | ab | 3.3 ± 0.44 | a–e | 5.2 ± 0.17 | ab | 4.7 ± 0.15 | abc |
| Parfianka | 3.2 ± 0.64 | ab | 1.9 ± 0.40 | c–i | 3.7 ± 0.24 | bcd | 3.4 ± 0.28 | a–f |
| Rose | 3.2 ± 1.02 | abc | 1.9 ± 0.76 | c–i | | | | |
| Sakerdze | 1.5 ± 0.75 | bc | 2.0 ± 0.41 | c–i | 2.5 ± 0.39 | def | 2.0 ± 0.68 | d–h |
| Salavatski | 2.6 ± 0.87 | bc | 2.3 ± 0.38 | a–i | 1.6 ± 0.41 | efgh | 1.9 ± 0.27 | efgh |
| Sin Pepe | 1.9 ± 0.66 | bc | 1.5 ± 0.45 | d–i | | | | |
| Sirenevyi | 2.2 ± 1.27 | bc | 0.6 ± 0.47 | ghi | | | | |
| Surh-Anor | **1.0 ± 0.54** | **bc** | 0.9 ± 0.48 | fghi | 3.0 ± 0.81 | cde | 3.2 ± 0.30 | b–f |
| Sweet | 1.6 ± 1.02 | bc | 2.3 ± 0.59 | a–i | 4.1 ± 0.42 | abcd | 4.0 ± 0.31 | abcd |
| Vkusnyi | 1.6 ± 0.60 | bc | 1.7 ± 0.34 | d–i | 1.8 ± 0.37 | efgh | 2.3 ± 0.44 | d–h |

Letters represent the differences among cultivars. In cases where more than 4 characters are present, a dash is used as a shorthand, e.g., a–i is abcdefghi.

## 2.2. Evaluation of Fruit Rot Severity under Natural Disease Pressure

Twelve or eighteen young fruit per cultivar were randomly selected and tagged in May 2021 and May 2022. Fruit was examined weekly for a period of eight weeks from May to July. When fruit rot appeared, the size of the rotted area (lesion size) on each fruit was manually measured and rounded to the nearest whole numbers in cm. Lesion size measurements were then converted to a 0 to 6 disease severity scale: 0 = no disease; 1 = lesions only occurring on the calyx; 2 = lesion of 1 or 2 cm; 3 = lesion of 3 or 4 cm; 4 = lesion of 5 or 6 cm; 5 = lesion 7 cm or greater; and 6 = the fruit had dropped from the tree due to fruit rot disease.

## 2.3. Preparation of Fungal Spore Suspension for Artificial Inoculation

Fungal isolate C30 for *C. gloeosporioides* was cultured on a potato dextrose agar (PDA) and incubated at (30 °C) for 12–15 days. A spore suspension was prepared by flooding the plates with 6 mL of autoclaved distilled water and scraping the agar surface. The spore suspension was then filtered through cheesecloth into a 50 mL Falcon tube. Spore density was determined using a hemacytometer and adjusted with autoclaved distilled water to a final suspension of $1 \times 10^5$ conidia/mL.

### 2.4. Inoculation of Fungal Spores to Open Flowers and Young Fruit on Plants in the Field

Artificial inoculation was achieved by applying 500 μL of $1 \times 10^5$ conidia/mL *C. gloeosporioides* inoculum into open hermaphroditic flowers that were at the anther dehiscence stage or the young fruitlet stage. Inoculated flowers and fruit were each enclosed in a mesh bag and a brown paper bag for 24 h to provide a high humidity environment for promoting fungal infection. After 24 h, the paper bag was removed, but the mesh bag was left in place to protect the inoculated fruit. In each growing season, 14 to 36 young fruit per cultivar were inoculated. For each cultivar, mock inoculations were performed on three to five fruit using sterile deionized water (SDW).

### 2.5. Measuring Lesion Size and Disease (Fruit Rot) Development after Artificial Inoculation of In-Planta Fruit the Field

All inoculated fruit were tagged and examined for fruit rot lesion size every week. The examinations continued for 8 weeks, from May into early July 2021 or mid-July 2022. Lesion measurements were then converted to a 0 to 6 ranking scale as described above.

### 2.6. Ranking of Fruit Rot Resistance Levels

Pomegranate cultivars were ranked for fruit rot resistance based on an empirical method that considered the average disease severity rating from both the natural infection and artificial inoculation over two years and the percent of fruit that had symptoms that were at level 5 or 6 on the above ranking scale. Only cultivars with complete data for both artificial inoculation and natural infection over the two years were ranked.

### 2.7. Re-Isolation of Fungal Pathogen from Inoculated Fruit

During the final three weeks of the evaluation, ten fruit showing symptoms of *Colletotrichum* fruit rot were collected at random from the field for a total of 30 fruit each year. Isolates were recovered from the fruit following the protocol of Xavier et al. (2019) [17]. After 10 days, the fungal cultures were visually examined to count the number of isolates that were *Colletotrichum*.

### 2.8. Evaluation of Detached Mature Fruit for Resistance to Fruit Rot

Mature fruit for many pomegranate cultivars were not available in Florida, so mature fruit were harvested from an experimental orchard at the USDA National Clonal Germplasm Repository (NCGR) in Davis, California, in the fall of 2022 and shipped to Florida. Seven available pomegranate cultivars with varying levels of fruit rot resistance were selected, including 'Afganski' (susceptible), 'Al-Sirin-Nar' (highly susceptible), 'Azadi' (highly resistant), 'Eversweet' (resistant), 'Fleishman' (resistant), 'Kazake' (highly susceptible), and 'Nikitski Ranni' (highly susceptible).

Fruit received from NCGR were first washed to remove any soil or debris, surface-sterilized by soaking in 0.0025% sodium hypochlorite for 30 min, and then air dried in the laboratory under ambient conditions. Individual surface-sterilized fruit were wounded to a 5 mm depth with a 3 mm diameter nail and sterilized with 70% ethanol after each use. Four wounds were made at equal distance apart around the center of each fruit. To each wound, 30 μL of $1 \times 10^5$ *C. gloeosporioides* spore inoculum was applied, allowing five minutes for the inoculum to absorb into the wound before sealing with petroleum jelly. Mock inoculations were made using sterile deionized water (SDW). Inoculated fruit were placed into a clear plastic container lined with damp paper towels and incubated for 14 days in a 24 °C growth chamber with a 12 h photoperiod. Starting six days after inoculation, fruit lesions were measured every two days. The detached fruit inoculation experiment was repeated four times, each time with four fruit per cultivar and four inoculated sites per fruit.

### 2.9. Statistical Analysis

Analysis of variance was conducted to test the disease responses of different pomegranate cultivars. Mean separation procedures among cultivars were conducted using a Tukey

HSD test. Pearson's correlation between infection methods and between years was tested using the cor.test function. All statistical analysis was performed in R (version 4.2.3) and the 'agricolae' [38] package was used for the HSD comparison.

## 3. Results

### 3.1. Fruit Rot under Natural Disease Pressure

Fruit rot began to appear on young fruit in May in both years and continued to enlarge as the fruit developed. Under natural infection, different patterns of disease development were observed over the 8 weeks in 2021 (Supplementary Figure S1). For the cultivars that were highly susceptible, disease progression happened even at the early fruit development stage and almost linearly, and rapidly increased over the season, especially as conditions for fungal growth improved with the beginning of the rainy season (Figure 3). For some of the more resistant cultivars such as 'Azadi' and 'Fleishman', disease progression happened at a much slower pace over the observation period and disease symptoms did not tend to expand on the fruit. A few cultivars showed little disease at the early phase but consistently experienced a sharp increase in disease symptoms later in the season when the conditions for fungal growth were more optimal. These cultivars included 'Eversweet', 'Girkanets', and 'Parfianka'.

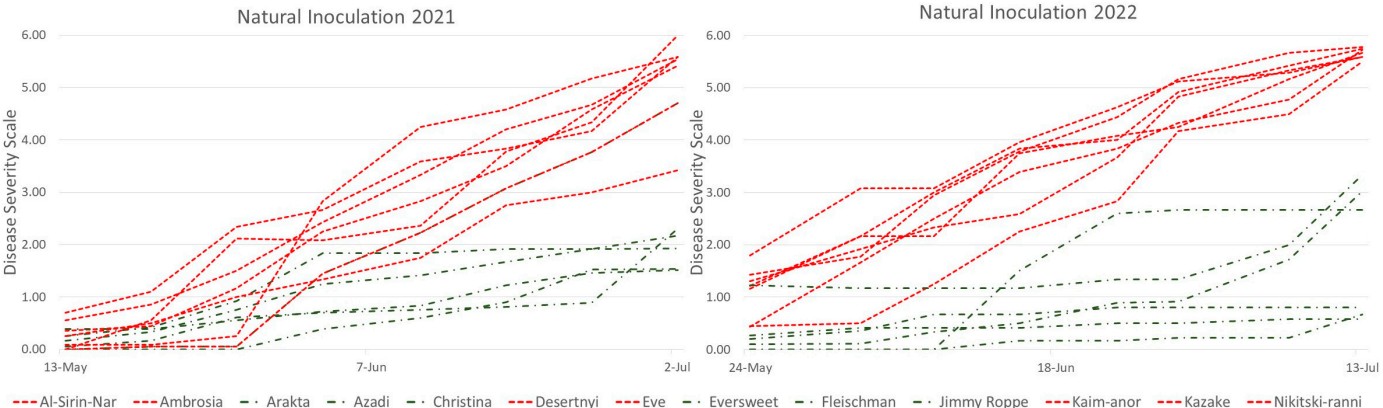

**Figure 3.** Average fruit rot progression in the most susceptible and resistant pomegranate cultivars over the 8 weeks of observation for naturally infected fruit during the 2021 and 2022 seasons.

At the end of the evaluation in 2021, the cultivars 'Azadi' (0.1), 'Arakta' (0.4), and 'Fleishman' (0.4) had the lowest level of disease severity, while the cultivars 'Kazake' (4.6), 'Desertnyi' (4.4), and 'Eve' (4.3) had the highest level of disease severity. In 2022, similar patterns of disease progression were observed as in 2021 (Figure 3; Supplementary Figure S2). At the end of the evaluation in 2022, 'Azadi' (0.3), 'Fleishman' (0.5), and 'Arakta' (0.8) had the lowest level of disease severity at the end of the evaluation, while 'Kazake' (5.3), 'Eve' (5.2), and 'Al-Sirin-Nar' (5.0) had the highest level of disease severity.

### 3.2. Fruit Rot after Artificial Inoculation

Under artificial inoculation during the 2021 season, fruit rot progressed linearly across the 8 weeks for all cultivars (Figure 4; Supplementary Figure S3). During the 2022 season, a similar trend was observed for all but five cultivars, including 'Eversweet' and 'Girkanets', which experienced a sharp increase in disease symptoms towards the end of the 8 weeks (Figure 4; Supplementary Figure S4).

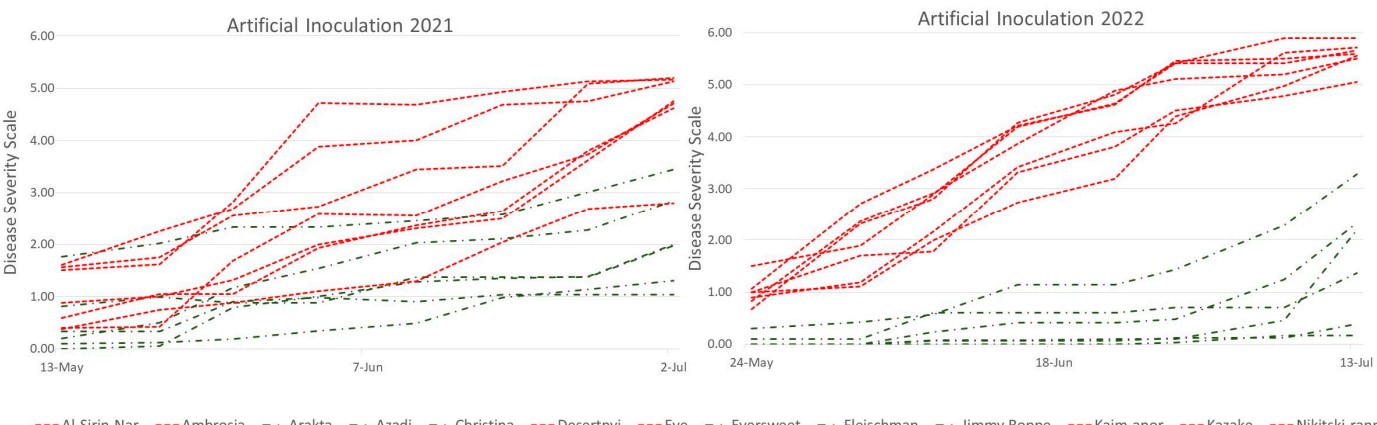

**Figure 4.** Average fruit rot progression of the most susceptible and resistant individuals over the 8 weeks of observation of artificially inoculated fruit for the 2021 and 2022 seasons.

There were statistically significant differences among cultivars in disease severity in both 2021 and 2022 (Table 1) for the artificially inoculated fruit. The cultivars 'Azadi' (0.1), 'Fleishman' (1.0), and 'Surh-Anor' (1.0) had the lowest level of disease severity in 2021, while the cultivars 'Kazake' (5.2), 'Molla Nepes' (4.4), and 'Ambrosia' (4.0) had the highest severity ratings. In 2022, the cultivars 'Azadi' (0.1), 'Christina' (0.2), and 'Eversweet' (0.7) had the lowest disease severity ratings in 2022, while 'Eve' (5.5), 'Al-Sirin-Nar' (5.3), and 'Kazake' (5.4) had the highest severity ratings.

At the end of the growing season, inoculated fruit were collected to re-isolate the pathogen. During the 2021 evaluation, out of the 30 fruits selected from the field, *Colletotrichum* was re-isolated from infected fruit tissue 16/30 times. During the 2022 evaluation, *Colletotrichum* was re-isolated from infected fruit tissue 18/30 times.

### 3.3. Ranking of Resistance Level and Correlation of Inoculation Methods and Years

Out of the 35 cultivars evaluated, 27 had data for all four categories over the two years and were included in the ranking of resistance. Of those 27 cultivars, one was highly resistant ('Azadi'), five were resistant, eight were moderately resistant, six were susceptible, and seven were highly susceptible (Table 2). The average percent of fruit that had a ranking of 5 or 6 at the end of the year was included for each class of phenotype, along with the average disease rating over the two years between the two infection methods.

**Table 2.** The 27 pomegranate cultivars that were included in ranking of resistance.

| Categories of Fruit Rot Resistance Level | Cultivars | % Fruit Rot Mean | Disease Rating Mean (0–6 Scale) |
|---|---|---|---|
| Highly Resistant | Azadi | 2 | 0.2 |
| Resistant | Arakta, Christina, Eversweet, Fleishman, Jimmy Roppe, | 25 ± 1 | 1.2 ± 0.15 |
| Moderately Resistant | Angel Red, Bala Miursal, Cranberry, Don Somner South, Sakerdze, Salavatski, Surh-Anor, Vkusnyi | 39 ± 2 | 2.2 ± 0.10 |
| Susceptible | Afganski, Girkanets, Grenada, Larkin, Parfianka, Sweet | 61 ± 2 | 2.8 ± 0.12 |
| Highly Susceptible | Al-Sirin-Nar, Ambrosia, Desertnyi, Eve, Kaim-anor, Kazake, Nikitski Ranni | 87 ± 2 | 4.4 ± 0.13 |

There was a significant positive correlation between the inoculation methods in both 2021 (r = 0.653, *p* < 0.0001) (Figure 5A) and 2022 (r = 0.876, *p* < 0.0001) (Figure 5B) and both

years (Figure 5C) (r = 0.794, *p* < 0.0001). There was also significant correlation between the two years (r = 0.694, *p* < 0.0001) (Figure 5D). These significant correlations suggest that the rankings of fruit rot resistance level among pomegranate cultivars in different years were consistent and under genetic control.

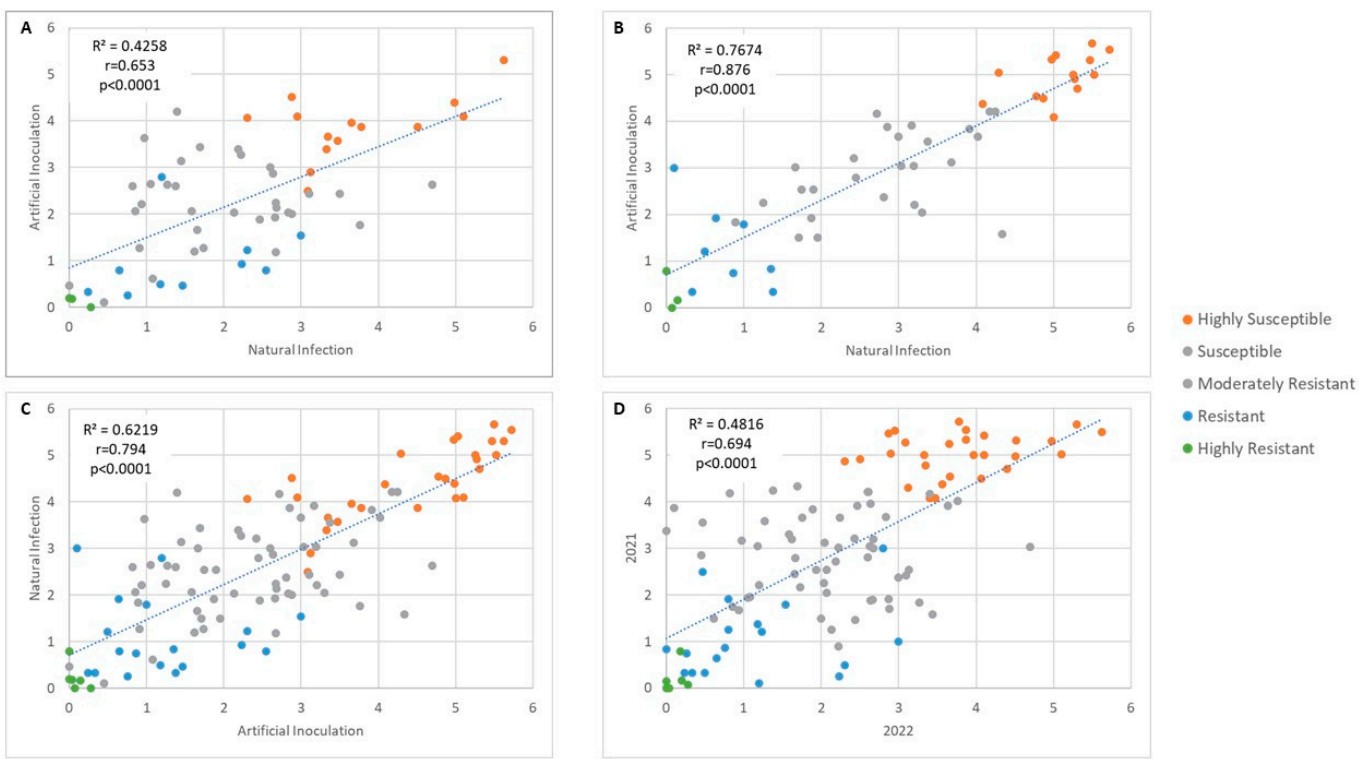

**Figure 5.** Correlation of disease severity rating between artificial inoculation and natural infection in 2021 (**A**), 2022 (**B**), and both years (**C**), and disease severity rating for both artificial inoculation and natural infection between years (**D**). Data points are color coded by ranking assigned in Table 2.

### 3.4. Detached Fruit Assay

There were significant cultivar differences in all four experiments, and overall, with 'Azadi' consistently having a smaller lesion diameter in comparison to all other cultivars (Figure 6 and Table 3). 'Afganski' and 'Kazake' were the most susceptible cultivars and had the largest lesions in all experiments except Experiment 2 (Table 3).

**Table 3.** Anthracnose fruit rot severity (lesion size) (diameter, in mm) for seven pomegranate cultivars after artificial inoculation of detached mature fruit in four replicated experiments. 'Fleishman' was not included in the fourth experiment due to a shortage of fruit available for inoculation. Letters represent the differences among cultivars. The standard error is included for each cultivar.

| Cultivar | Experiment 1 | | 2 | | 3 | | 4 | | Average |
|---|---|---|---|---|---|---|---|---|---|
| Afganski | $22.1 \pm 1.9$ | a | $20.6 \pm 3.8$ | abc | $20.3 \pm 1.6$ | a | $12.8 \pm 2.8$ | abc | 19.6 |
| Al-Sirin-Nar | $17.8 \pm 1.1$ | a | $21.2 \pm 2.8$ | ab | $9.4 \pm 2.0$ | bc | $8.3 \pm 1.9$ | bcd | 14.9 |
| Azadi | $7.6 \pm 1.7$ | b | $8.9 \pm 1.9$ | c | $4.7 \pm 0.9$ | c | $3.0 \pm 0.0$ | d | 6.0 |
| Eversweet | $21.7 \pm 4.5$ | a | $27.5 \pm 4.0$ | a | $8.3 \pm 1.5$ | bc | $4.0 \pm 0.5$ | cd | 15.5 |
| Fleishman | $13.8 \pm 2.2$ | ab | $18.8 \pm 3.7$ | abc | $5.1 \pm 1.2$ | bc | | | 12.7 |
| Kazake | $22.0 \pm 1.9$ | a | $12.0 \pm 1.7$ | bc | $20.8 \pm 1.4$ | a | $18.5 \pm 1.5$ | a | 18.4 |
| Nikitski Ranni | $15.3 \pm 2.1$ | ab | $23.8 \pm 3.0$ | a | $11.7 \pm 2.1$ | b | $12.9 \pm 1.9$ | ab | 15.4 |

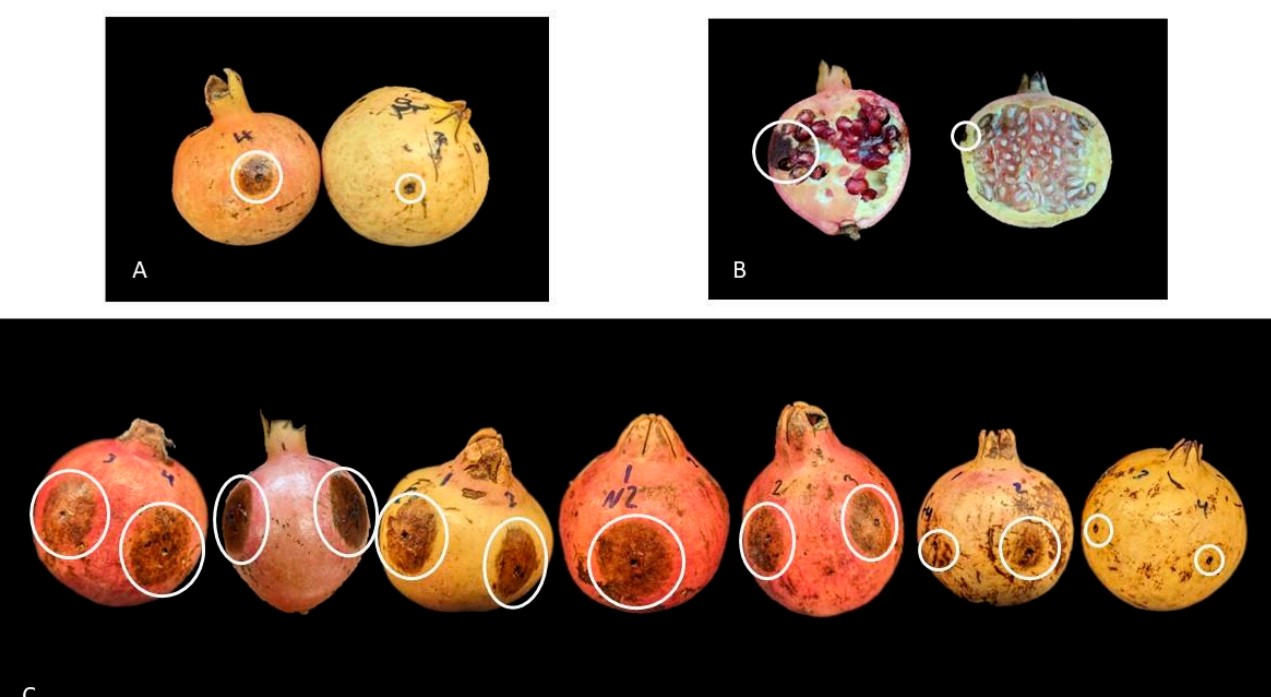

**Figure 6.** Lesion size comparison among detached mature fruit of pomegranate cultivars. (**A**,**B**) compare two representative cultivars, 'Kazake' on the left and 'Azadi' on the right, 14 days post-inoculation, externally and internally. (**C**) shows lesion sizes of seven cultivars, from the largest average lesion size to the smallest lesion size ('Afganski', 'Kazake', 'Eversweet', 'Nikiski Ranni', 'Al-Sirin-Nar', 'Fleishman', and 'Azadi'). Lesions are indicated with white circles.

## 4. Discussion

Resistance to anthracnose fruit rot is a much-needed trait for pomegranates, especially in the southeastern United States and other subtropical regions in the world where the environmental conditions are ideal for the growth and spread of the causal fungal pathogen *C. gloeosporioides*. Our results indicate that there are very valuable sources of genetic resistance to this pathogen in the current USDA pomegranate germplasm collection as well as in local heirloom varieties. 'Azadi' seems to be the most promising cultivar and has strong resistance to *C. gloeosporioides*, and a handful of other cultivars also have useful resistance.

In many studies investigating disease resistance in different cultivars, repeatability and differences between years, artificial inoculation, and natural infection, as well as inoculation methods, are often problematic [39–41]. In-field evaluations were further complicated in this experiment by the conducive environment for pathogen spread and the high number of pathogens that infect pomegranates present in Florida [19]. While re-isolation of *Colletotrichum* from infected fruit in the field was possible, multiple other fungal pathogens were also isolated from infected tissues collected from the field. This pathogen pressure meant that many of the mock inoculated fruit showed symptoms of fungal infection due to natural infection occurring, particularly in cultivars with higher susceptibility levels. Despite the many factors that could affect infection in the field study portion, there was statistical significance as well as a strong magnitude of correlation when looking at the correlation of different inoculation methods within years and between years. This would suggest that the rankings of resistance for cultivars were consistent, regardless of the inoculation method (artificial or natural infection) or year, despite the varying environmental effects.

Evaluation of the fruit over the season seems to suggest that in a few cultivars, there was an increase in susceptibility as the fruit matured. For example, 'Eversweet' and 'Girkanets' showed an increase in lesion size late in the season as the fruit approached

maturity. These results suggest that a fruit's age may play a role in its resistance to fruit rot. Among the other six cultivars tested in the detached fruit study, the results were consistent with field data. Such consistency may suggest a possibility of screening pomegranate cultivars for fruit rot resistance by inoculating detached fruit in the laboratory rather than inoculating open flowers or young fruit in the field, which is much more challenging, time-consuming, and complicated. Disease resistance screening based on detached fruit would be much easier to implement and better to control environmental variables such as temperature and relative humidity, resulting in an enhanced selection efficiency.

'Arakta' has been evaluated for disease resistance in other studies. Joshi et al. 2014 [32] assessed it for fruit rot resistance, and Jayalakshmi et al. (2015) [33] for leaf spot resistance. Both studies rated it as highly susceptible to *Colletotrichum*. However, in our study, we found 'Arakta' to be resistant with low disease incidence in both artificial inoculations as well as natural infections. These differences may be due to factors such as different pathogen populations, the environment, or interactions among these factors.

Interestingly, of the cultivars that showed resistance or strong resistance to anthracnose fruit rot in this study, all but 'Arakta' have a yellow to light pink peel and light pink arils. In other crops, disease resistance has been found to increase as fruit gain more red color. These pomegranate cultivars do not seem to follow that trend [42,43]. It will be interesting to find out whether fruit peel or aril color is associated with fruit rot resistance in pomegranate. If such an apparent association does exist in pomegranate, it may provide an easy-to-use, visual marker for screening pomegranate breeding populations for anthracnose fruit rot resistance. On the other hand, such an association may slow down the progress for developing new anthracnose-resistant cultivars with deep red fruit peel and arils, which are more popular than yellow to light pink peel or light pink arils among consumers.

Four of the resistant cultivars ('Eversweet', 'Fleishman', 'Christina', and 'Jimmy Roppe') originated within the United States, with the first two from California and the last two from northern Florida/Southern Georgia. 'Azadi' and 'Arakta', on the other hand, originated in Turkmenistan and India, respectively. It is extremely interesting that 'Azadi', originating from a region where anthracnose is not a common problem, has evolved strong resistance to anthracnose. These cultivars may be worth being evaluated for resistance to local *C. gloeosporioides* isolates in other tropical and subtropical regions where anthracnose fruit rot is a major disease.

These anthracnose-resistant cultivars have been crossed with other cultivars with other desirable traits including high yield and appealing external and internal color to develop new cultivars with anthracnose resistance, high yield, and superior quality. Resistant cultivars with different origins have also been intercrossed with the hope that their progeny may have stronger or broader-spectrum resistance than their parents. Genome and transcriptome sequencing is underway to identify candidate genes and develop molecular markers for this disease resistance trait so that large breeding populations can be screened more efficiently using molecular markers.

In recent years, a number of highly destructive diseases have been reported in pomegranate [20,22,24,44–48]. Resistance to diseases has become a much-needed trait for new pomegranate cultivars, and more research has been devoted to find sources of disease resistance, understand their genetic and molecular mechanisms, and develop new tools to incorporate them into new cultivars. For example, Kumari and Ram (2015) [49] evaluated 63 cultivars for resistance to *Coniella granati* and found six cultivars with moderate resistance to this pathogen. Jabnoun-Khiareddine et al. (2018) [50] evaluated nine cultivars and their response to *Coniella granati*; however, they found that all cultivars showed some level of susceptibility. Mincuzzi et al. (2020) [51] revealed that the cultivar 'Wonderful' had a higher resistance to *Coniella granati* than 'Mollar de Elche' and an up-regulation of genes associated with chitinase, phenylalanine ammonia-lyase (PAL), and peroxidase genes as well as phenolic compounds. Priya et al. (2016) [52] discovered five pomegranate genotypes with resistance to the highly destructive disease bacterial blight (BB) caused by *Xanthomonas axonopodis* pv. *punicae*. Kumar et al. (2021) [53] identified three pomegranate accessions

with strong tolerance to BB and found that resistant accessions had an up-regulation of phenylalanine ammonia-lyase, callose synthase-3 (CS3), chitinase, pathogenesis-related protein-1 (PR1), and pathogenesis-related protein-10 (PR10) genes. A number of simple sequence repeat (SSR) markers have been associated with BB resistance or tolerance in pomegranate [54]. Overall, our knowledge of disease resistance traits and the availability of genetic, genomic, and molecular tools for these traits in pomegranate are very limited compared to what have been developed in other fruit crops. It is expected that as more genomic resources become available [55–58], they will accelerate the genetic improvement of disease resistance in this important crop.

## 5. Conclusions

Anthracnose is extremely destructive to pomegranate fruit, particularly in subtropical regions where it is warm, humid, and rainy, and the environmental conditions are ideal for the disease. This study represents a crucial breakthrough toward the development of new *Colletotrichum*-resistant pomegranate cultivars. 'Azadi' and five other cultivars that have demonstrated resistance or high resistance to *Colletotrichum* fruit rot merit further horticultural tests for potential commercial production in these regions. These sources of resistance are playing a pivotal role in developing new pomegranate cultivars for Florida and elsewhere where *Colletotrichum* anthracnose fruit rot is prevalent.

Disease resistance has become a very important objective in pomegranate breeding programs. Our studies indicate that pomegranate germplasm from different countries may harbor highly valuable sources of resistance for major diseases. Preserving such germplasm, including local or heirloom varieties, deserves more attention as climate change and highly destructive diseases become more prevalent in the pomegranate-producing areas in the world.

To enhance our understanding of this vital resistance trait and efficiently utilize it in breeding efforts, future studies need to investigate the inheritance of fruit rot resistance and its genetic relationship with other traits including fruit skin colors and phenolic compounds, identify and locate the gene loci for the observed resistance in 'Azadi' and other cultivars, and develop molecular and genomic selection tools.

Our study reveals the existence of resistance, even strong resistance, to destructive diseases in the pomegranate germplasm that originated from different parts of the world. This exciting finding suggests tremendous potential to improve pomegranate disease resistance and protect this vital crop against the devastating impacts of anthracnose.

**Supplementary Materials:** The following supporting information can be downloaded at: https://www.mdpi.com/article/10.3390/horticulturae9101097/s1, Table S1: Cultivars and their Origin, Figure S1: Disease progression of natural infection 2021. Figure S2: Disease progression of natural infection 2022. Figure S3: Disease progression of artificial inoculation 2021. Figure S4: Disease progression of artificial inoculation 2022.

**Author Contributions:** A.S.: formal analysis, investigation, methodology, project administration, writing—original draft, and writing—review and editing. J.M.C.: conceptualization, funding acquisition, and writing—reviewing and editing. G.E.V.: methodology, resources, and writing—reviewing and editing. J.M.: resources. C.H.: resources. Z.D.: conceptualization, funding acquisition, project administration, resources, supervision, and writing—reviewing and editing. All authors have read and agreed to the published version of the manuscript.

**Funding:** This project was supported by the USDA Agricultural Marketing Service (AMS) MultiState Specialty Crop Block Grant through the California Department of Food and Agriculture project number 19-1043-002-SF, the AMS Specialty Crop Block Grant through the Florida Department of Agriculture and Consumer Services (FDACS) Specialty Crop Block Grant Program (Contract #00099169), and the former Florida Pomegranate Association.

**Data Availability Statement:** Not applicable.

**Acknowledgments:** The authors are grateful to J. Velte, C. Willborn, and S. Ledon for their assistance with field data collection; J. Jones, G. Bowman, K. Druffel, and G. Garcia Garcia for maintaining the pomegranate orchard; and C. Weistein, R. Bonsteel, J. Alexander, and D. Bice for providing pomegranate plants and their support.

**Conflicts of Interest:** The authors declare no conflict of interest.

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
