# Peer review of "Pomegranate Cultivars with Diverse Origins Exhibit Strong Resistance to Anthracnose Fruit Rot Caused by Colletotrichum gloeosporioides, a Major Disease in Southeast United States"

_horticulturae, doi:10.3390/horticulturae9101097_

Round 1

Reviewer 1 Report

This article evaluated the anthracnose resistance in 35 cultivars of pomegranate through a 2-year field study. The results are important for breeding resistant pomegranate varieties in the future and for controlling anthracnose fruit rot in pomegranate production.

However, I have read numerous articles about anthracnose infection of plants or fruit. Then, what is the innovation of your research work?

The Abstract section. I suggest that the author should reduce the background description and present the research content of this paper as much as possible.

3.4. Re-Isolation of Pathogen from Field Material. The authors should present the results of the isolation of specific anthracnose pathogens.

Author Response

This article evaluated the anthracnose resistance in 35 cultivars of pomegranate through a 2-year field study. The results are important for breeding resistant pomegranate varieties in the future and for controlling anthracnose fruit rot in pomegranate production.

However, I have read numerous articles about anthracnose infection of plants or fruit. Then, what is the innovation of your research work?

Response and revisions:       Anthracnose fruit rot is not a new disease, and similar research has been reported in some other fruit crops. But this is the first time in the world that multiple sources of strong resistance to anthracnose fruit rot have been discovered in pomegranate cultivars that originated from areas far apart. These sources of resistance are playing a critically important role in our effort to develop new anthracnose-resistant pomegranate cultivars as a major tool for managing anthracnose fruit rot, the economically most important disease in the southeast United States that often causes a total loss of pomegranate crop. The resistance we have identified showed stable expression in the field and on mature detached fruit, indicating high value for improving pomegranate for resistance to this destructive disease.

Several diseases, including blackheart disease, cause huge economic losses in the commercial production of pomegranate in the world, especially in subtropical areas. However, very limited information is available in the world to use genetic resistance to manage these diseases. Our pioneering study will lead the pomegranate breeding community toward the development of long-term, cost-effective, and environmentally sound solutions to disease management.

The Abstract section. I suggest that the author should reduce the background description and present the research content of this paper as much as possible.

Response and revisions: We have reduced the words on the research background and increased words on research results and new findings.                    

3.4. Re-Isolation of Pathogen from Field Material. The authors should present the results of the isolation of specific anthracnose pathogens.

Response and revisions:       We have included results from our re-isolation experiments in both years.

We have made extensive revisions.  These revisions can be seen in the attached Word document that tracks the revisions.  Please let us know if you have additional questions.  Thank you very much for your time and comments!

Reviewer 2 Report

The study was performed on fungal pathogen Colletotrichum and to determine how pomegranate resist against it. Although the study is interesting, however its look an incomplete work  because the authors selected only few cultivars while they did not provide sufficient information/experiment how some cultivars resist more than others.

Quality of english may be improved.

Author Response

Methods:         Can be improved

Response:       Have improved it. Please find details in our revised manuscript.

Results:           Must be improved

Response:       Have improved it. Please find details in our revised manuscript.

Conclusions:   Must be improved

Response:       Have improved it. Please find details in our revised manuscript.

Comments and suggestions for authors

The study was performed on fungal pathogen Colletotrichum and to determine how pomegranate resist against it. Although the study is interesting, however its look an incomplete work because the authors selected only few cultivars while they did not provide sufficient information/experiment how some cultivars resist more than others.

Quality of English may be improved.

Response and revisions:       The main focus of this paper was germplasm evaluation to identify resistant cultivars. As mentioned in the conclusion, further work is being done to determine mechanisms of resistance in these cultivars. We are working on this and have created segregating populations and are doing genome re-sequencing to investigate the resistance mechanisms.  This work will take a number of years. Meanwhile pomegranate researchers can use our new findings to advance pomegranate breeding for disease resistance, which is a dire need for pomegranate growers.

We have improved the manuscript and made it more clear and more succinct. Hope our revisions are acceptable.

We have made extensive revisions.  These revisions can be seen in the attached Word document that tracks the revisions.  Please let us know if you have additional questions.  Thank you very much for your time and comments!

Reviewer 3 Report

Dear authors

The following modifications are required:

Title

ü  In general, this section is poorly written. It is written simply. This section should include the most important findings from this study. As a result, this section should be improved.

ü  The aim is not stated correctly.

ü  This section should contain details about the experimental design as well as the components or levels of factors.

ü  The authors should provide some measurement values for the parameters under consideration.

Keywords

ü  The terms used in the forming of the title should not be used as the keywords, so the content of keywords should be changed

Introduction

ü  This section is generally poorly written. It is written in a straightforward manner. The authors should provide some information about the detrimental effects of Colletotrichum species

ü  The authors should give some lines about the knowledge gap which their research has covered along with the hypothesis statement

ü  Also, the authors should provide a novelty statement at the end. What new things authors have done or correlated in this research compared to old ones?

ü  The general and specific aim should be inserted

Materials and Methods

ü  Some climatological data of the region should be added

Results

ü  In Tables 1 and 3, the authors should define the different letters

ü  The standard error or deviation should be added to the data of Tables 1 and 3

Discussion

ü  This section is poorly written and should be improved

ü  The authors should provide some information about the results of correlation between the natural and artificial infection.

ü  The authors should mention some information about the variability of the response of different cultivars

Conclusion

ü  This section is excessively long and contains information that is unrelated to the conclusion. The authors should be arranged

ü  The authors should include some information about future works in this field.

moderate correction should be conducted

Author Response

Introduction:   Must be improved

Response:       Have improved it. Please find details in our revised manuscript.

References:     Must be improved

Response:       Have improved it. Please find details in our revised manuscript.

Research design:         Must be improved

Response:       Have improved it. Please find details in our revised manuscript.

Methods:         Must be improved

Response:       Have improved it. Please find details in our revised manuscript.

Results:                       Must be improved

Response:       Have improved it. Please find details in our revised manuscript.

Conclusions:   Must be improved

Response:       Have improved it. Please find details in our revised manuscript.

Reviewer comments and suggestions for authors

Title

ü  In general, this section is poorly written. It is written simply. This section should include the most important findings from this study. As a result, this section should be improved.

ü  The aim is not stated correctly.

ü  This section should contain details about the experimental design as well as the components or levels of factors.

ü  The authors should provide some measurement values for the parameters under consideration.

Response and revisions:       We have a new title to better describe project and the main results. We have edited abstract to show the study’s aims, include more details on the experimental design and how measurements were performed.

Keywords

ü  The terms used in the forming of the title should not be used as the keywords, so the content of keywords should be changed

Response and revisions:       Key words were changed so that words that were in the title were not also included in the key words.  

Introduction

ü  This section is generally poorly written. It is written in a straightforward manner. The authors should provide some information about the detrimental effects of Colletotrichum species

ü  The authors should give some lines about the knowledge gap which their research has covered along with the hypothesis statement

ü  Also, the authors should provide a novelty statement at the end. What new things authors have done or correlated in this research compared to old ones?

ü  The general and specific aim should be inserted

Response and revisions:       We have revised the Introduction section and included a new Figure to provide more information on the effects of Colletotrichum species in pomegranate, described the knowledge gap in pomegranate, and shown the novelty of the experiment and its importance of identifying sources of resistance for the future of pomegranate production.

Materials and Methods

ü  Some climatological data of the region should be added

Response and revisions: We added Figure 2 to demonstrate the climatic data of the area the experiment took place.

Results

ü  In Tables 1 and 3, the authors should define the different letters

ü  The standard error or deviation should be added to the data of Tables 1 and 3

Response and revisions: We have updated the caption for both tables, defined the letters and the differences amongst cultivars, and included the standard errors in both Table 1 and 3.

Discussion

ü  This section is poorly written and should be improved

ü  The authors should provide some information about the results of correlation between the natural and artificial infection.

ü  The authors should mention some information about the variability of the response of different cultivars

Response and revisions:  We have revised and improved this section by including the correlation between the natural and artificial infection, stating a clear strong correlation between the two infection methods as well as between years, and showing relatively consistent ranking of anthracnose rot resistance or susceptibility despite environmental differences between the two years.

Conclusion

ü  This section is excessively long and contains information that is unrelated to the conclusion. The authors should be arranged

ü  The authors should include some information about future works in this field.

Response and revisions: We have revised and shortened this section and included future research we are doing or have planned to do. 

Quality of English Language: Moderate corrections should be conducted.

Response and revisions: We have polished the manuscript and improved clarity by consulting with native English-speaking professors. Hope now the quality of the English language is acceptable.       

We have made extensive revisions.  These revisions can be seen in the attached Word document that tracks the revisions.  Please let us know if you have additional questions.  Thank you very much for your time and comments!

Reviewer 4 Report

The subject of the manuscript is very interesting and has applied rather than scientific value. In this regard, the publication in a journal of such a high level is very doubtful. This study lacks a scientific approach, perhaps genetic analysis or the search for determinants of pomegranate resistance to this pathogen. And so the study comes down to simply assessing the degree of damage to different varieties. In addition, there are a number of other remarks:

1. Of course, many varieties were studied, but the use of 2-3 trees for analysis is not enough for a statistically significant analysis and interpretation of the results. The result could be influenced by the direct location of a particular tree. Therefore, the results obtained are highly questionable.

2. There is no table that lists all varieties with their place of growth, with information on resistance from the breeder, the number of trees from each variety.

3. The discussion section is small and unpresentable. The presented varieties have a description, the breeder suggests or checks varieties for resistance to pathogens, and these parameters were not taken into account for all varieties.

4. It is necessary to search for the genetic component of resistance, and not just a statement of the fact of resistance. This study lacks a scientific approach and the use of modern methods.

Author Response

Introduction:   Must be improved

Response:       Have improved it. Please find details in our revised manuscript.

References:     Must be improved

Response:       Have improved it. Please find details in our revised manuscript.

Research design:         Not applicable

Methods:         Can be improved

Response:       Have improved it. Please find details in our revised manuscript.

Results:           Must be improved

Response:       Have improved it. Please find details in our revised manuscript.

Conclusions:   Not applicable

Reviewer comments and suggestions for authors

The subject of the manuscript is very interesting and has applied rather than scientific value. In this regard, the publication in a journal of such a high level is very doubtful. This study lacks a scientific approach, perhaps genetic analysis or the search for determinants of pomegranate resistance to this pathogen. And so the study comes down to simply assessing the degree of damage to different varieties. In addition, there are a number of other remarks:

Response:       We absolutely agree that genetic analysis and finding of the determinants of disease resistance are very important, but these analyses will be based on discovery of pomegranate genotypes or cultivars with varying degrees of disease resistance susceptibility or resistance. When such plant materials and knowledge of disease resistance levels are not available, it would be impossible to conduct desired genetic analyses or discovery of the gene loci for the resistance. The research findings described in this manuscript have filled a critical knowledge gap and provided essential experimental materials for pomegranate scientific research and horticulture. These newly discovered sources of resistance can have remarkable values for pomegranate breeding, genetics, genomics, production, etc. The availability of these materials have enabled us to develop segregating populations for inheritance studies, genome mapping, transcriptome profiling, etc., those types of genetic analyses important for science.

  1. Of course, many varieties were studied, but the use of 2-3 trees for analysis is not enough for a statistically significant analysis and interpretation of the results. The result could be influenced by the direct location of a particular tree. Therefore, the results obtained are highly questionable.

Response and revisions:       The experimental unit in this study was not individual pomegranate trees, but individual pieces of fruit. For each cultivar in each year and for each inoculation method, 12 to 36 pieces of fruit were used. Over two years, 26 to 54 pieces of fruit per cultivar were evaluated for resistance to the disease.  

  1. There is no table that lists all varieties with their place of growth, with information on resistance from the breeder, the number of trees from each variety.

Response and revision:        Unlike other major agronomic or horticultural crops, information on cultivar’s disease resistance and area of production is not available in pomegranate. This is a huge knowledge gap for commercial production and scientific research, which was why this study was initiated. We have included a supplementary table to show the origin and fruit color of the cultivars used in this study, as these factors may play some roles in the cultivar resistance to anthracnose. The number of trees for each cultivar was described in the manuscript, so it was not repeated in the supplementary table.  

  1. The discussion section is small and unpresentable. The presented varieties have a description, the breeder suggests or checks varieties for resistance to pathogens, and these parameters were not taken into account for all varieties.

Response and revision:        We have revised and strengthened the Discussion section by discussing other instances of fungal resistance explored in pomegranate. We have included additional information (origin, fruit color) in the supplementary table.

  1. It is necessary to search for the genetic component of resistance, and not just a statement of the fact of resistance. This study lacks a scientific approach and the use of modern methods.

Response:       The availability of these disease-resistant materials has enabled us to develop segregating populations for inheritance studies, genome mapping, transcriptome profiling, etc. These newly initiated studies will allow us to find the genetic factors that determine the disease resistance reported in this study. However, pomegranate seedlings need 2-4 years before they can flower and produce fruit for phenotyping for anthracnose fruit rot resistance. Thus, it will take at least 2 to 4 years before such knowledge could be available. In the meantime, breeders are eager to use disease-resistant cultivars as parents to make new breeding populations and develop new resistant cultivars, which makes timely publication of this manuscript and our findings very important to the researcher community and pomegranate industry. 

We have made extensive revisions.  These revisions can be seen in the attached Word document that tracks the revisions.  Please let us know if you have additional questions.  Thank you very much for your time and comments!

Round 2

Reviewer 2 Report

I recommend the paper to be accepted for publication in its current form. 

Author Response

I recommend the paper to be accepted for publication in its current form. The following sections can be improved.

Response:       Thank you.

Introduction:   Can be improved

Response:       Revised this section to stress the importance of research on pomegranate and disease resistance.

References:     Can be improved

Response:       Added more references related to disease resistance and genomic resources in pomegranate.

Results:           Can be improved

Response:       Revised this section to improve clarity.

Reviewer 3 Report

The authors have been addressed all comments

moderate correction is needed

Author Response

The authors have been addressed all comments. The following sections can be improved. Moderate correction is needed.

Response:       Thank you.

Introduction:   Can be improved

Response:       Revised the Introduction to stress the importance of research on pomegranate and disease resistance.

References: Can be improved

Response:       Added references related to disease resistance and genomic resources in pomegranate.

Research design: Can be improved

Response:       Revised this section to improve clarity.

Methods:         Can be improved

Response:       Made revisions to improve clarity.

Results: Can be improved

Response:       Revised this section to improve clarity.

Conclusions: Can be improved

Response:       Made revisions to strengthen our conclusions.

Reviewer 4 Report

As the authors said, this cultivar is quite new in the United States. The cultivars used in the study are of foreign origin. Therefore, perhaps, our own varieties should be created for the USA. However, this point of view is not really well considered. Many countries around the world have been growing pomegranate for centuries (Azerbaijan, India, China, Spain, Iran, Turkey and others). Researchers from these countries have published a large number of articles, including in international English-language publications. However, the authors of the manuscript did not properly analyze the experience of foreign researchers. Such an important culture in the world and only 32 analyzed literary sources (most of which do not reflect world experience). The same or similar cultivars have not been analyzed for their degree of resistance in similar climatic conditions. A thorough analysis has not been carried out, clear recommendations have not been given, and there are no clear conclusions. Therefore, the value of this work for the global community of scientists is not clear.

Also, the description of the study itself (materials and methods) also requires improvement. Where is table 1 in the manuscript? So how many cultivars took part in the experiment: 35, 27, 12 or 18. Why specific varieties are selected for inoculation. That is why the purpose of the experiment is not clear, what the hypothesis is, what conclusions were expected and what were the results.

A very unfinished study, also not recommended for publication.

Author Response

Comments and suggestions for authors:

As the authors said, this cultivar is quite new in the United States. The cultivars used in the study are of foreign origin. Therefore, perhaps, our own varieties should be created for the USA. However, this point of view is not really well considered (#1). Many countries around the world have been growing pomegranate for centuries (Azerbaijan, India, China, Spain, Iran, Turkey and others). Researchers from these countries have published a large number of articles, including in international English-language publications. However, the authors of the manuscript did not properly analyze the experience of foreign researchers (#2). Such an important culture in the world and only 32 analyzed literary sources (most of which do not reflect world experience) (#3). The same or similar cultivars have not been analyzed for their degree of resistance in similar climatic conditions (#4). A thorough analysis has not been carried out, clear recommendations have not been given, and there are no clear conclusions (#5). Therefore, the value of this work for the global community of scientists is not clear.

Response to Comment #1:      Thank you for pointing out that “our own varieties should be created for the USA.” We totally agree. This was the reason we screened these cultivars. We have used the newly discovered resistant cultivars as breeding parents, made multiple new crosses, and developed new breeding populations. Please find these changes in lines 358 – 363. These resistant cultivars identified in this study will play a pivotal role in developing new cultivars in the USA.

Response to Comment #2:      A great deal of research has been done on pomegranate, but mostly on botany, breeding, physiology, horticulture, postharvest physiology, pathology, etc. So far, limited research has been done on identifying sources of resistance to major fungal diseases. We did a pretty extensive literature search and didn’t find much more literature on this topic. We did find a good amount of literature on pomegranate resistance or susceptibility to bacterial blight, which is caused by Xanthomonas axonopodis pv. purnicae. Research on bacterial blight has been a good example for us to follow. We have revised the Introduction and Discussion sections to include more literature on these aspects.

Response to Comment #3 – Only 32 references included:                We have included more than 20 more references that are related to the work reported here.                    

Response to Comment #4 - “The same or similar cultivars have not been analyzed for their degree of resistance in similar climatic conditions”:  This is where the value of this work is for the world researcher community, especially in other tropical and subtropical regions where pomegranates are being trialed. We hope other researchers can use our findings and begin to test these resistant cultivars with their Colletotrichum isolates. Please see additional revisions in Lines 363-365.

Response to Comment #5 – “A thorough analysis has not been carried out, clear recommendations have not been given, and there are no clear conclusions”:  As we pointed out in the Discussion section, a lot of research can be done to understand the genetic and molecular mechanisms for the identified resistance, to locate the genetic loci, to identify candidate genes, to develop molecular markers, etc., but our current conclusions seem to be clear – These cultivars are resistant to anthracnose and can be useful as breeding parents or as cultivars for production.

Also, the description of the study itself (materials and methods) also requires improvement. Where is table 1 in the manuscript? So how many cultivars took part in the experiment: 35, 27, 12 or 18. Why specific varieties are selected for inoculation. That is why the purpose of the experiment is not clear, what the hypothesis is, what conclusions were expected and what were the results. A very unfinished study, also not recommended for publication.

Where is table 1 in the manuscript?

Response:       Table 1 is located in Lines 255 – 260.

How many cultivars took part in the experiment 35, 27, 12 or 18.   

Response:       There were 35 cultivars evaluated in 2021, however during the 2022 season due to production issues for some cultivars because of a late freeze, only 27 cultivars were available for evaluation. This is why only 27 cultivars were included in Table 2. This is described in Lines 127-128, and 273.

Why were specific varieties selected for inoculation?

Response:       All varieties that had available fruit were inoculated in the field; no varieties were chosen for in-field testing. For detached fruit evaluation, the seven cultivars were chosen to represent varying levels of fruit rot resistance. Also, these were the cultivars that were available from NCGR. This is outlined in Lines 189 - 190.

What is the hypothesis? What conclusions were expected? What were the results?

Response:       We have revised the text to indicate our hypothesis and expected conclusions. Please see the text in Lines 94-96.  The results are described in Lines 401 – 403.

Introduction:   Must be improved

Response:       Made revisions to stress the importance of research on pomegranate and disease resistance.

References:     Must be improved

Response:       Added more references related to disease resistance and genomic resources in pomegranate to reflect more pomegranate research done worldwide.

Research design:         Must be improved

Response:       Made revisions to clarify research design.

Results:           Must be improved

Response:       Made revisions to improve clarity.

Conclusions:   Must be improved

Response:       Made revisions to strengthen our conclusions.